# STABILIZE CONTINUAL LEARNING WITH HYPER-SPHERICAL REPLAY

## ABSTRACT

Neural networks face catastrophic forgetting of previously learned knowledge when training on new task data. While the field of continual learning has made promising progress in reducing this forgetting, recent work has uncovered an interesting phenomenon: existing techniques often exhibit a sharp performance drop on prior tasks during the initial stages of new task training, a phenomenon known as the "stability gap." This phenomenon not only raises safety concerns but also challenges the current understanding of neural network behavior in continual learning scenarios. Motivated by this discovery, we revisit two fundamental questions in continual learning: 1) Is the past learned knowledge within deep networks lost abruptly or gradually? and 2) Is past learned knowledge ever completely erased? Our analysis reveals that abrupt forgetting occurs not only in the final fully connected layer but also permeates the feature space and most layers, sparing only the earliest layers. Alarmingly, a single gradient update can severely disrupt the learned class structure. We identify degenerate solutions in the softmax cross-entropy loss as a major contributing factor, with memory samples exhibiting higher feature norms compared to new samples. To address these issues, we propose Adaptive Angular Replay (AAR), a simple yet effective approach that learns features in hyperspherical space using feature and weight normalization. Angular ER demonstrates a strong ability to preserve class structure during task transitions. Additionally, we introduce an adaptive scaling strategy to further mitigate the stability gap and improve overall accuracy.

## 1 INTRODUCTION

Machine learning has increasingly relied on training large models on static datasets to achieve impressive results, often surpassing human capabilities in a wide range of tasks. However, these tasks are typically confined and static after deployment, reflecting a key limitation of neural network optimization: the assumption of independent and identically distributed (iid) training and testing data. In real-world scenarios, data is dynamic, continuously evolving with new information arriving at an unprecedented rate, often violating the iid assumption. As a result, neural networks are prone to "catastrophic forgetting" (CF) (French, 1999; Delange et al., 2021), where models abruptly lose previously learned knowledge when exposed to new, non-iid data. In such cases, learning agents face the challenge of absorbing new information efficiently. To address these challenges, fields like continual learning (CL) and lifelong learning have gained significant attention, focusing on reducing the impact of catastrophic forgetting while adapting to changing data distributions.

Research efforts to mitigate catastrophic forgetting have led to various promising solutions, with replay-based methods achieving state-of-the-art performance (Hadsell et al., 2020; Wang et al., 2021). Despite their success, recent studies have revealed an unexpected phenomenon: while rehearsal-based continual learning techniques reduce forgetting, they still exhibit significant performance drops during the initial phase of training on new tasks. This temporary performance decline, followed by recovery, is termed the "stability gap"(De Lange et al.). Though transient, the stability gap introduces potential risks to continual learning systems and underscores the need for a deeper understanding of how neural networks behave in CL settings.

**Fundamental Open Questions** In this paper, we explore a critical yet unresolved question in the literature: *Does the knowledge embedded in a neural network degrade abruptly or gradually during*

*continual learning?* Empirical studies generally suggest that as more tasks are introduced (Delange et al., 2021; Mai et al., 2022), performance on prior tasks deteriorates, leading to the assumption of gradual knowledge loss. However, these evaluations are typically conducted after each task is fully trained. When performance is monitored at every gradient step, recent findings on stability gap suggest a different dynamic: performance on previous tasks drops sharply in the early stages of training on a new task and only gradually recovers afterward. This abrupt drop raises the possibility of sudden knowledge loss during training. Nevertheless, a drop in task performance does not necessarily indicate a complete loss of knowledge across the entire network. Previous works have identified a task-recency bias in the final fully connected (FC) layer (Wu et al., 2019; Mai et al., 2022; Zhao et al., 2020), which can significantly affect the final performance. One hypothesis is that the stability gap is driven by this bias in the FC layer. Some supporting evidence for this is that when visualizing the training trajectories in the loss landscape, the network parameters drift slowly from low-loss regions to higher-loss areas as training progresses (Verwimp et al., 2021; Zhang et al., 2022). In summary, although it has been demonstrated there is abrupt loss of task's performance, whether the core network experiences abrupt or gradual knowledge loss—and to what extent of knowledge retention or loss within the deeper network layers—remains unclear.

**Key findings**. In this work, we investigate the network's change dynamics during task transitions to answer these questions. Our findings reveal that the stability gap extends beyond the final FC layer, affecting the network's internal representations. Notably, we show that the stability gap persists even when using a Near-Class-Mean classifier instead of cross-entropy classifier. Centered Kernel Alignment (CKA) analysis reveals abrupt changes in representations in later network layers, while earlier layers experience more gradual and subtle shifts. Crucially, we observe that the class structure in the feature space can be entirely disrupted by just a single gradient step, underscoring the intensity of knowledge loss in the network's deeper layers.

**Proposed Solutions**. To mitigate this abrupt loss of class structure and address the stability gap, we identify degenerate solutions in the softmax cross-entropy loss as a key factor. This degeneration leads to much higher feature norms for memory samples compared to the new samples. To address this issue, we propose a simple but effective solution called Adaptive Angular Replay (AAR), which promotes learning in hyper-spherical space using feature and weight normalization. Angular ER preserves the class structure more effectively than prior methods. Additionally, to further reduce the stability gap and improve overall accuracy, we introduce an adaptive scaling strategy that complements Angular replay. Together, these methods significantly enhance the performance of continual learning systems by preserving knowledge more effectively throughout training.

**Contributions** Our contributions are as follows:

- We provide several insights into knowledge retention and loss in non-stationary data settings: 1) The stability gap extends beyond the final FC layer and affects the entire network and feature space. 2) There is a complete loss of class structure in the feature space during task transitions. 3) Knowledge loss in later layers is abrupt, whereas in early layers, it is more gradual.

- We identify degenerate solutions in the cross-entropy loss that result in higher feature norms, contributing to the loss of class structure.

- We propose Adaptive Angular Replay, a simple and effective solution to mitigate the stability gap by learning features in hyperspherical space, complemented by an adaptive scaling factor to further enhance performance.

## 2 RELATED WORK

**Continual learning**: We consider the online continual learning setting with a non-stationary (potentially infinite) stream of data $\mathcal{D}_t$: at each time step $t$, the continual learning agent receives an incoming batch of data samples $\mathcal{B}_t = \{\mathbf{x}_i, y_i\}_{i=1,...,|\mathcal{B}_t|}$ that are drawn from the current data distribution $\mathbb{P}(\mathcal{D}_t)$. The period of time where the data distribution stays the same is often called a *task* or *experience* in the continual learning literature. An abrupt change in the data distribution occurs when the task changes. The standard objective during training is to minimize the empirical risk on

all the data seen so far:

$$\min_\theta \mathcal{R}(\theta) = \min_\theta \frac{1}{\sum_t |\mathcal{B}_t|} \sum_t \sum_{\mathbf{x},y \in \mathcal{B}_t} \mathcal{L}\left(f_\theta(\mathbf{x}), y\right), \tag{1}$$

with loss function $\mathcal{L}$, the CL network function $f$, and its associated parameters $\theta$.

**Stability Gap**: (De Lange et al.) identified stability gap. This phenomenon is further discussed and studied in the context of pre-trained large language model (Guo et al., 2024) and in the incremental Learning of Homogeneous Tasks (Kamath et al., 2024). Our work focuses on conventional continua learning settings with non-stationary data.

**Forgetting mitigation techniques**. Continual learning algorithms address catastrophic forgetting in three main ways: replay-based methods (Chaudhry et al., 2018; Aljundi et al., 2019) store and replay past samples to mitigate forgetting; regularization-based methods (Rebuffi et al., 2017; Li & Hoiem, 2017) use regularization losses to encourage retention of past knowledge; architecture-based methods Mallya & Lazebnik (2018); Serra et al. (2018) separate parameters for different tasks to avoid interference.

**Hyperspherical embedding**. Hyperspherical embedding has gained significant attention in various machine learning domains. The concept of hyperspherical prototypical networks Mettes et al. (2019) is proposed for few-shot learning and demonstrates improved performance by constraining embeddings to lie on a hypersphere. In the context of continual learning, the effectiveness of hyperspherical embedding remains under-explored. In particular, when employing the Cross-Entropy loss in continual learning, it is typically computed based on dot similarity between the feature vector and prototype vector. Our work investigates how and why hyperspherical features are particularly useful for reducing the stability gap in continual learning. We explore the implications of using hyperspherical embeddings and analyze their impact on the stability and performance of continual learning models.

## 3 ANALYSIS: THE NETWORK BEHAVIOR AT TASK TRANSITION

Understanding how and why the stability gap occurs is crucial not only for practical applications but also as a scientifically intriguing phenomenon that can deepen our understanding of network learning behaviors in the context of non-stationary data distributions. We aim to use this phenomenon as a lens to explore the processes of information loss and retention during the learning of new information. To this end, we present a series of analyses focused on the behavior of the network during task transitions.

### 3.1 REVISIT A SIMPLE BASELINE: NEAREST CLASS MEAN CLASSIFIER

The cross-entropy classifier is the most widely used option in continual learning. In a CE classifier, the model's final layer is a fully connected layer with weight matrix $W \in \mathbb{R}^{D \times N}$, where $N$ is the number of classes. A well-known phenomenon in continual learning, referred to as *task-recency bias* or *biased fully connected layer* (Wu et al., 2019; Zhao et al., 2020), occurs when the logits output and norm of the weights corresponding to new classes becomes significantly higher than that of old classes. This bias is believed to contribute to catastrophic forgetting. A common explanation attributes this bias to class imbalance, as the exemplar set storing past data is often small, meaning the number of samples for new classes typically exceeds that for old classes.

To explore whether the final FC layer causing the stability gap, we revisit the simple baseline of the Nearest-Class-Mean (NCM) classifier, which has been employed in several continual learning studies as a method to reduce forgetting (Rebuffi et al., 2017; Mai et al., 2021). Unlike the cross-entropy classifier, NCM does not rely on a fully connected layer for predictions but instead uses learned features to compute class prototypes from memory samples. Inference is then performed based on the distance between the input and the nearest class prototype. More specifically, it computes a prototype vector for each class observed so far, $\mu_1, ... \mu_c$ where $\mu_c = \frac{1}{|P_c|} \sum_{p \in P_c} \varphi(p)$ is the average feature vector of all exemplars for a class $c$. It also computes the feature vector of the image that should be classified and assigns the class label with most similar prototype:

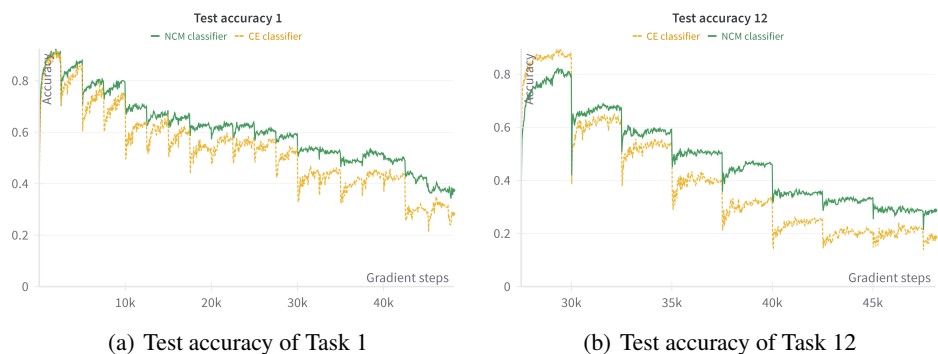

(a) Test accuracy of Task 1          (b) Test accuracy of Task 12

Figure 1: Nearest-Class-Mean classifier vs. cross-entropy classifier. The experiment is conducted by splitting CIFAR100 into sequential 20 tasks)

$$y^* = \underset{y=1,\dots,t}{\arg\min} \left\| \varphi(x) - \mu_y \right\|. \tag{2}$$

While NCM has been shown to mitigate forgetting and enhance overall accuracy at the end of training, its effect on the stability gap remains unclear. We compare the performance of continual evaluation using CE and NCM classifiers in Figure 1. Our results demonstrate that NCM significantly reduces the stability gap compared to the CE classifier. However, it's noteworthy that a gap persists even with the NCM classifier.

Specifically, the test accuracy for the first task exhibits sharp "spikes," where performance drops dramatically during task transitions before gradually recovering. This observation suggests that the stability gap is not solely attributable to the final fully connected (FC) layer, but also involves changes in the underlying feature representations.

## 3.2 LOSS AND RETENTION OF CLASS STRUCTURE

We investigate how task transitions affect learned features, particularly in two aspects: 1) the extent to which class structure is disrupted in the feature space, and 2) how the forgetting-recovery behavior propagates through the network layers.

**The extent of knowledge loss**. To address the first question, we visualize feature representations at three key points in training: a) before training on a new task, b) after a *single gradient step* on the new task, and c) after the new task's training is complete. As shown in Fig 2, we observe a complete loss of class structure after just one gradient step (Fig 2 b). This surprising result raises the question of whether past knowledge is fully erased in the whole network or if some degree of information is retained.

**The scope of knowledge loss in the network**. To further investigate how information is lost or retained during task transitions, we turn to Centered Kernel Alignment (CKA), a neural network representation similarity measure. CKA and other related algorithms provide a scalar score (between 0 and 1) determining how similar a pair of (hidden) layer representations are, and have been used to study many properties of deep neural networks.

$$\text{CKA}(\text{X}, \text{Y}) = \frac{\text{HSIC}\left(\text{XX}^{\text{T}}, \text{YY}^{\text{T}}\right)}{\sqrt{\text{HSIC}^{\text{T}}\text{XX}^{\text{T}}, \text{XX}^{\text{T}})}\sqrt{\text{HSIC}^{\text{SI}}\text{YY}^{\text{T}}, \text{YY}^{\text{T}})} \tag{3}$$

To assess how the model evolves during the training of a new task, we compare the hidden representations at each gradient step to those of the model before the new task training begins. Specifically, in Equation 3, $\text{X} = \phi_t^L$ represents the activation at a particular layer for all memory data during gradient step $t$. Correspondingly, $\text{Y} = \phi_0^L$ represents the activation at the same layer for all memory data before the training of the new task commences.

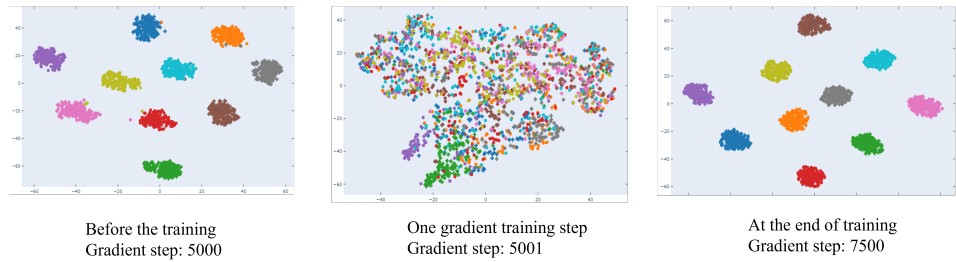

Before the training
Gradient step: 5000

One gradient training step
Gradient step: 5001

At the end of training
Gradient step: 7500

Figure 2: Loss of class structure in the feature representation: T-sne visualization of representation of memorized samples during the training.

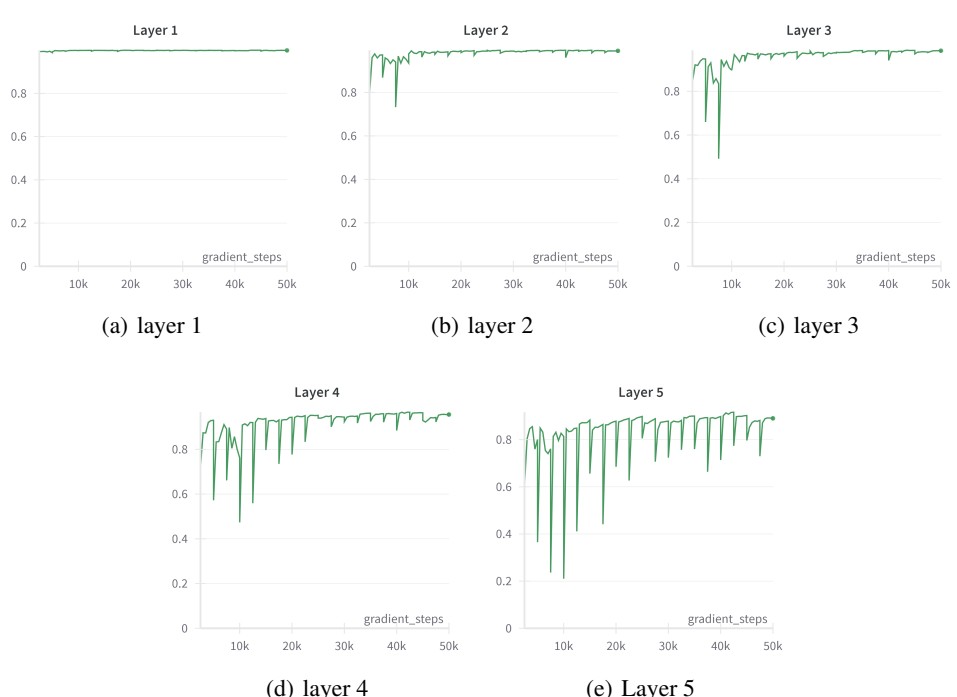

(a) layer 1        (b) layer 2        (c) layer 3

(d) layer 4        (e) Layer 5

Figure 3: Sudden knowledge loss in the backbone of network measured by CKA: hidden representations changes happen during task transition except the early layers

Figure 3 reveals that abrupt changes occur predominantly in deeper layers (layers 4 and 5) throughout the learning process. The most significant alterations to representations take place during the first gradient step, followed by a period of partial recovery. In middle layers (2 and 3), sudden changes are observed only for the initial few tasks. As the model encounters more tasks, these abrupt shifts become less pronounced. The shallow layer (layer 1) exhibits no sudden changes.

## 4 METHODS: ADAPTIVE HYPERSPHERICAL REPLAY

### 4.1 HYPERSPHERICAL REPLAY TO MAINTAIN THE CLASS STRUCTURE

In this section, we investigate the perspective of loss function and analyze how CE loss may be problematic for continual learning and shows that a simply modification can reduce the stability challenge significantly.

**The effect of degenerate solutions in softmax cross-entropy loss**. Equation 4 gives an equivalent form of CE loss. Based on this, we have Equation 5, which suggest that as long as the feature can

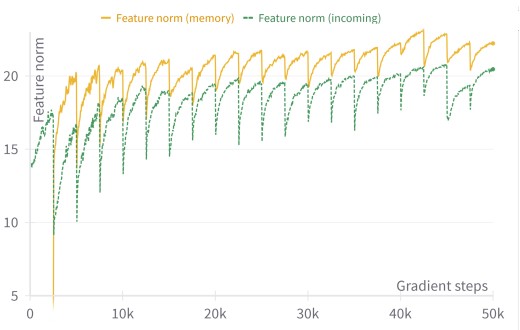

Figure 4: Feature norm disparity: The feature norm of memory samples are significantly higher than that of new samples.

be correctly classified the loss can be trivially further minimized by increasing feature norm. However, increasing feature norm does not necessarily making the feature more discriminative. These degnerate solutions with high feature norms does not influence IID setting much as it influences all training samples. However, this is particularly problematic for continual learning. As the memory samples are trained repeatedly by the model, while the new samples are never seen by the model at the start of training. Thus, we hypothesize that degeneracy in CE loss can lead to a large disparity in the feature norm of memory samples and new samples.

$$\mathcal{L}_{\text{CE}} = \log\left(1 + \sum_{i \neq y} \exp\left(\boldsymbol{w}_i^\top \boldsymbol{\phi} - \boldsymbol{w}_y^\top \boldsymbol{\phi}\right)\right) \tag{4}$$

$$\lim_{\|x\| \to \infty} \mathcal{L}_{\text{CE}} = \begin{cases} 0 & \text{if} \quad \forall i \neq y, \boldsymbol{w}_y^\top \boldsymbol{\phi} > \boldsymbol{w}_i^\top \boldsymbol{\phi} \\ +\infty & \text{if} \quad \exists i \neq y, \boldsymbol{w}_y^\top \boldsymbol{\phi} < \boldsymbol{w}_i^\top \boldsymbol{\phi} \end{cases} \tag{5}$$

Informally, we summarize this the relationship between feature norm and CE loss in the following claim.

*Claim 1* (informal): In continual learning scenarios, given sufficient computational iterations with cross-entropy loss, the feature norms of memory samples consistently and substantially exceed those of new task samples.

We verify this claim empirically in Figure 4. The norm of features of memory samples is significantly higher than that of the new samples. As the softmax score is linearly related to feature norm. An immediate problem arising from this feature norm difference is that it leads to a large disparity between the softmax scores and the loss of new samples and memory samples, which increases the stability gap.

**Angular Similarity**. To avoid the effect of degeneracy in CE loss, we propose to train the CE loss in the hypersphere. In particular, we write CE loss in the form of cosine similarity. By assume a zero bias vector and normliza the weight vector and feature vector to be 1. We have angular CE in Equation 7.

$$\mathcal{L}_{CE} = -\log\left(\frac{e^{\boldsymbol{W}_{y_i}^T \boldsymbol{\phi}_i + b_{y_i}}}{\sum_j e^{\boldsymbol{W}_j^T \boldsymbol{\phi}_i + b_j}}\right)$$

$$= -\log\left(\frac{e^{\|\boldsymbol{W}_{y_i}\|\|\boldsymbol{\phi}_i\|\cos\left(\theta_{y_i,i}\right) + b_{y_i}}}{\sum_j e^{\|\boldsymbol{W}_j\|\|\boldsymbol{\phi}_i\|\cos\left(\theta_{j,i}\right) + b_j}}\right) \tag{6}$$

$$L_{angular} = -\frac{1}{N}\sum_{i=1}^{N} \log \frac{e^{s(t)\cos\theta_{y_i}}}{e^{s(t)\cos\theta_{y_i}} + \sum_{j=1,j\neq y_i}^{n} e^{s(t)\cos\theta_j}} \tag{7}$$

where $S(t)$ is a scaling factor.

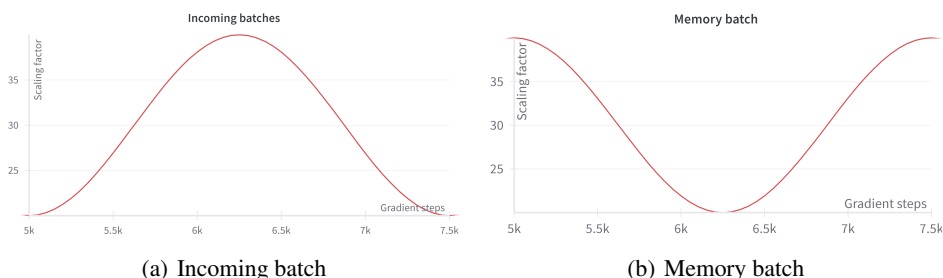

(a) Incoming batch  (b) Memory batch

Figure 5: Adaptive scaling factor: Fade-in-Fade-out schedule.

Figure 6 (a) shows that angular CE successfully maintains the class structure during task transition despite its simplicity.

## 4.2 FADE-IN-FADE-OUT: ADAPTIVE SCALING

$$\frac{\partial \ell_{ce}}{\partial h_j} = \begin{cases} s(t)(p_j - 1) \leq 0, & q_j = q_y = 1 \\ s(t) \times p_j \geq 0, & q_j = 0 \end{cases} \tag{8}$$

**The effect of scaling factor**. The scaling factor plays a crucial role in controlling the "sharpness" of the learning signal and the dynamics of gradients. As shown in Equation 8, it linearly affects the magnitude of the entire gradient. Moreover, increasing the scaling factor (i.e., lowering the temperature) makes the output probability distributions $p_j$ more extreme (closer to 0 or 1), potentially leading to larger gradient differences across class logits. Conversely, decreasing the scaling factor (i.e., raising the temperature) results in a more uniform distribution, reducing these differences and making the model's predictions more uncertain or diverse.

**Adaptive Scaling**. To enhance the stability of the continual learning process, we propose an adaptive schedule for adjusting the scaling factor, as shown in Equation 10. This approach employs distinct scaling factors for memory batches and incoming batches, implemented in two phases (see Figure 5):

1. "Fade-in" Phase: During the initial stage, the scaling factor for incoming batches begins at a low value $s_{min}$ and gradually increases over time to $s_{max}$. This allows the model to slowly adapt to new information.

2. "Fade-out" Phase: Towards the end of training, we gradually decrease the scaling factor for incoming batches while simultaneously increasing it for memory batches. This strategy helps mitigate forgetting of previously learned information.

The transitions between these phases and the rate of scaling factor adjustment are controlled by a cosine function, ensuring smooth and continuous changes throughout the training process.

$$s_t^{inc} = s_{\min} + \frac{1}{2}\left(s_{\max} - s_{\min}\right)\left(1 + \cos\left(\frac{T_{\text{cur}}}{T}2\pi - \pi\right)\right) \tag{9}$$

$$s_t^{mem} = s_{\min} + \frac{1}{2}\left(s_{\max} - s_{\min}\right)\left(1 + \cos\left(\frac{T_{\text{cur}}}{T}2\pi\right)\right) \tag{10}$$

where $T_{\text{cur}}$ denotes the current gradient step, and $T$ represents the total number of gradient steps in the task training process.

## 5 EXPERIMENTS

### 5.1 EXPERIMENT SETUP

Three continual learning benchmarks are used in the experiments: Seq-CIFAR100-20 randomly splits the 100 classes of CIFAR100 into 20 sequential tasks. Each task contains five classes. Seq-MiniImageNet-10 randomly splits the 100 classes in mini-ImageNet (Vinyals et al., 2016) dataset

Table 1: Stability gap evaluation with worse case accuracy. * indicate the difference is statically significant comparing AAR with NCM.

| Dataset | Seq-CIFAR100 | Seq-Mini-ImageNet | CLRS |
|---|---|---|---|
| CE | $6.9 \pm 0.5$ | $8.5 \pm 0.5$ | $25.9 \pm 1.3$ |
| NCM | $13.8 \pm 0.7$ | $18.1 \pm 0.6$ | $26.6 \pm 0.8$ |
| ACE | $20.5 \pm 1.3$ | $9.9 \pm 0.3$ | $30.5 \pm 1.6$ |
| Angular | $30.7 \pm 1.2$ | $18.0 \pm 0.8$ | $39.1 \pm 1.0$ |
| AAR | $\mathbf{32.8^*} \pm 0.3$ | $\mathbf{18.2} \pm 1.3$ | $\mathbf{40.5^*} \pm 1.1$ |

Table 2: Final accuracy in three continual learning benchmarks.* indicates the performance difference is statistically significant based on t-test analysis.

| Dataset | Seq-CIFAR100 | Seq-Mini-ImageNet | CLRS |
|---|---|---|---|
| CE | $37.3 \pm 0.9$ | $33.6 \pm 0.6$ | $30.0 \pm 0.6$ |
| NCM | $44.9 \pm 0.8$ | $35.4 \pm 0.6$ | $36.2 \pm 0.9$ |
| ACE | $43.1 \pm 0.9$ | $35.9 \pm 0.6$ | $31.0 \pm 0.5$ |
| Angular | $43.6 \pm 0.4$ | $36.4 \pm 0.7$ | $42.3 \pm 1.3$ |
| AAR | $\mathbf{45.4^*} \pm 0.3$ | $\mathbf{36.4} \pm 1.3$ | $\mathbf{42.9^*} \pm 1.1$ |

into 10 tasks. CLRS25-NC is a real-world remote sensing dataset (Li et al., 2020). It contains 25 land cover classes, which are splitter into 5 tasks. Each tasks contains 5 classes.

We use a ResNet-18 for all datasets following (Mai et al., 2021; Aljundi et al., 2019). Single-head evaluation is employed with a shared final layer trained for all the tasks. We employ augmentation of random cropping and flipping and a memory size of 2000. The batch size for incoming data and memory data are both 50. The learning rate is 0.1. All the experimental results we present are averages of three runs.

A common metric is the end accuracy after training on $T$ tasks. Using $f_t$ to indicate the version of the model after the t-th overall training iteration, the accuracy of evaluation task $E_k$ at this iteration is denoted as $A(E_k, f_t)$. The end accuracy after $N$ tasks is defined as

$$\text{end-acc}_t = \frac{1}{N} \sum_{k=1}^{k=N} A(E_k, f_t)$$

We measure the stability gap following De Lange et al.. The stability gap is measured by worse-case accuracy instead of average accuracy, as follows.

$$\text{wc-acc}_t = \frac{1}{k} \mathbf{A}(E_k, f_t) + \left(1 - \frac{1}{k}\right) \text{min-acc}_{T_k} \tag{11}$$

where min-acc gives a worst-case measure of how well knowledge is preserved in previously observed tasks. More specifically, the average minimum accuracy (min-ACC) at current training task $T_k$ as the average absolute minimum accuracy over previous evaluation tasks $E_i$ after they have been learned:

$$\text{min-acc}_{T_k} = \frac{1}{k-1} \sum_{i}^{k-1} \min_{n} \mathbf{A}(E_i, f_n), \forall t_{|T_i|} < n \leq t \tag{12}$$

where the iteration number n ranges from after the task is learned until current iteration t.

## 5.2 RESULTS

As shown in Table 2, techniques including nearest-class-mean classifier, ACE ACE Caccia et al. (2021) and the proposed adaptive angular replay can all significantly improve the worse-case accuracy, with AAR achieving the largest improvement. Moreover, AAR can maintain the overall performance and enhance the overall performance, especially in the case of a real-world remote sensing datasets.

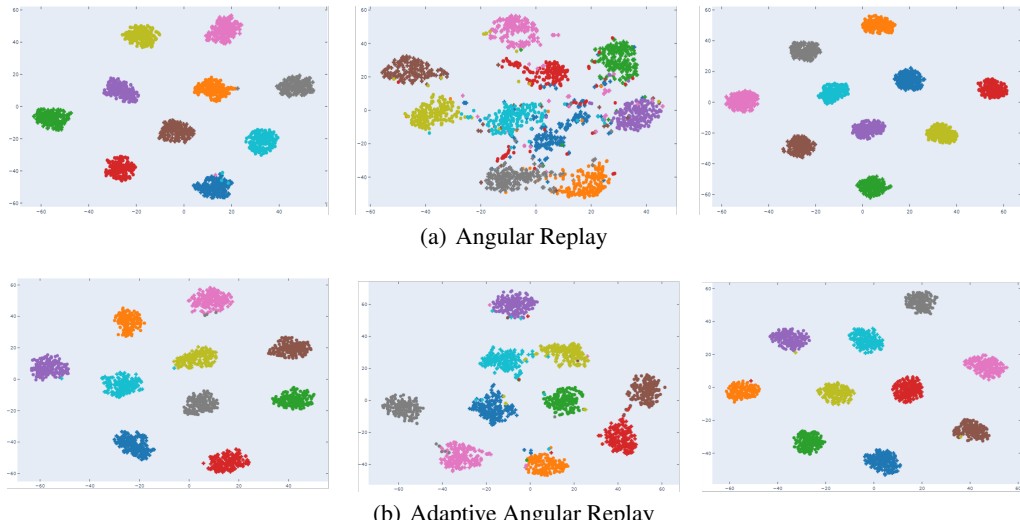

(a) Angular Replay

(b) Adaptive Angular Replay

Figure 6: Class structure retention in Angular replay and Adaptive Angular Replay. Tsne visualization of memory samples before the training of task 2 (left), after a single gradient step of the new task (middle), and at the end of training (right)

### 5.3 ABLATION STUDIES

We conduct an ablation study to study the effect of learning features in hypersperirical space and the effect of fade-in-fade-out scaling schedule. Figure 2 shows that using employing angular replay can help maintain the class structure. Compared to using a static scaling schedule, the proposed "fade-int-fade-out" strategy can further reduce the stability gap and maintain the class structure.

## 6 CONCLUSION

Rehearsal-based methods play a central role in fighting catastrophic forgetting when learning from non-stationary data streams. The phenomenon of stability gap raise question on current understanding of how and why rehearsal mitigates forgetting. Our analysis on the internal workings of network knowledge retention and loss reveals that 1) the stability gap is not confined to the final fully connected layer but affects the entire network and feature space and 2) there is a complete disruption of class structure in the feature space during task transitions, which can occur after just a single gradient step. To address the stability challenge in continual learning, we have developed Adaptive Angular Experience Replay (AAR), a novel approach that promotes learning in hyperspherical space. By using feature and weight normalization, Angular ER effectively mitigates the stability gap and preserves class structure more efficiently than existing methods. Furthermore, our proposed adaptive scaling strategy complements Angular ER, further reducing the stability gap and improving overall accuracy in continual learning systems.

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
