# OpenReview forum: "Stabilize continual learning with hyperspherical replay"
_ICLR.cc/2025/Conference — ICLR 2025 Conference Withdrawn Submission_

### Official Review · Reviewer_tRHL · 2024-11-02

**Soundness:** 2
**Presentation:** 2
**Contribution:** 2
**Rating:** 3
**Confidence:** 4

**Summary:**

The paper address a problem in Continual Learning approaches, in which past learned knowledge is abruptly erased by a single gradient step on a new task, leading to a forgetting of past learned patterns.

The authors proposed a method which works on different aspects of such phenomena by integrating multiple components, and evaluated it against different well established approaches.

**Strengths:**

The paper correctly identifies the phenomenon that lead to the abrupt forgetting of past learned knowledge. The paper is easy to follow and the approach justified.

**Weaknesses:**

Overall, the paper is easy to follow, but it requires some reworks in its structure, as well as regarding the relation between the claims and the literature, which is outdated. Specifically, here is a non-comprehensive list of weakness, errors, or suggested modifications:

- Eq 3 missing parenthesis; HSIC not defined. Additionally, I believe that this approach must be translated into numerical results to help compare the proposed approach against others.

- Forgetting mitigation techniques section lacks newer papers that addressed not only the forgetting but also the stability gap (e.g. [FI, LODE, SSIL]). In general, the paper lacks references to the recent literature

- Claim 1 is too strong, and it must be circumscribed to a more defined scenario. I suggest removing it

- Equation 8 is badly placed and not integrated in the text flow

- ACE written twice at beginning of section 5.2

- All figures from 1 to 4 and associated text are out of place. Firstly, the details about the training regime (dataset, task splitting, training approach, and others) are missing, secondly, these can be considered results and cannot be used to justify the approach, since it leads to a circular thesis. I suggest moving them in the experimental section and rework sections 3.1 and 3.2

- angular similarity similar loss (6-7) have been already used in CL [CM]. I believe this paper is worth citing along with [FI], which address the stability gap by fixing the classifiers.

- Section 5.3 does not contain ablations studies, but only references to figure 3, which was introduced to justify the method. Such section should contain an extensive study about the approach (e.g. what happens if you remove a part of your method) to validate its components.

- It is not clear which metric have you used to evaluate the results (tables 1 and 2). In general, the results section is chaotic and lacks of proper analysis of the results obtained. Additionally, the proposed method must be compared to more state of the art approaches

[FI] F. Pernici, M. Bruni, C. Baecchi, F. Turchini and A. Del Bimbo, "Class-incremental Learning with Pre-allocated Fixed Classifiers," 2020 25th International Conference on Pattern Recognition (ICPR), Milan, Italy, 2021, pp. 6259-6266, doi: 10.1109/ICPR48806.2021.9413299.
[CM] Pomponi, Jary, Simone Scardapane, and Aurelio Uncini. "Centroids Matching: an efficient Continual Learning approach operating in the embedding space." Transactions on Machine Learning Research (TMLR), 2022.
[CGC] Pomponi, Jary, Alessio Devoto, and Simone Scardapane. "Cascaded Scaling Classifier: class incremental learning with probability scaling." arXiv preprint arXiv:2402.01262 (2024).
[LODE] Y.S. Liang and W.-J. Li. Loss decoupling for task-agnostic continual learning. In Thirty-
seventh Conference on Neural Information Processing Systems, 2023
[SSIL] H. Ahn, J. Kwak, S. Lim, H. Bang, H. Kim, and T. Moon. Ss-il: Separated softmax for
incremental learning. In Proceedings of the IEEE/CVF International conference on computer
vision, pages 844–853, 2021.

**Questions:**

1. the feature norm increase probably because the model makes room for newer classes, and thus must increment the gap between the past one and the newly added one. What do you think about it?
2. why do you evaluate the features of the model instead of the produced distribution?
3. what happens if you normalize the features of the last layer while training? Is the norm of the rehearsal samples contained?
4. a recent paper [CGC] studied the same problem faced in this paper, and came to interesting conclusions. What do you think about the relations between the presented paper and the aforementioned one?

---

### Official Review · Reviewer_M2S3 · 2024-11-03

**Soundness:** 2
**Presentation:** 3
**Contribution:** 2
**Rating:** 3
**Confidence:** 5

**Summary:**

The paper studies the problem of class incremental learning and analyses the abrupt changes on the representation during class incremental learning sequence. The paper proposes to normalize the features and weights of the classifiers (few papers have suggested that before) and to adaptively scale the probability distribution of replay samples and new samples differently. Experiments on 3 task sequences show improvements of the proposed approach.

**Strengths:**

The insights presented in the paper and the follow is good.

The scaling seems to have a positive effect on the performance

**Weaknesses:**

The paper contribution and analysis is based on one experiment under one setting, it is unclear that this is the case for different class incremental tasks.

 While I enjoyed the way the contributions were presented, relying on simply on figures from a random step doesn't provide much of an evidence on the soundness of the insights and the solution.
For example, it is not stated for what replay buffer size this is analysis is done, and for which network, Even te CKA analysis is done before stating which network is used and evidence is made on layer 4 without stating layer 4 of what.
Even the memory is mentioned before introducing that replay is deployed,

Why only a certain gradient step is shown? i.e., 5001?

CKA is  not introduced properly and without a reference, what is HSIC in eq3.

The equations are not well presented, some terms are undefined, for example in equation 5, Wi is not defined before, and it is stated in the lim of |x| but shouldn't it be \phi? even \phi is not defined.
eq4,5 are presented with no bias term but later it is stated that the bias will be omitted.

NCM is discussed but it is not clear with which loss function?

**Questions:**

The hyper angular replay was used before in https://arxiv.org/pdf/2104.05025 Eq2,  could the authors comment on that?
How does the conclusions and the method behave under different replay size? and different architecture, Resnet is quite outdated..
How many gradients steps are there per task?

---

### Official Review · Reviewer_azYW · 2024-11-06

**Soundness:** 2
**Presentation:** 2
**Contribution:** 1
**Rating:** 3
**Confidence:** 4

**Summary:**

This paper investigates knowledge loss during task transitions in continual learning. The key findings are that early layers degrade gradually, but deeper layers abruptly lose learned knowledge. To address this, the paper proposes Adaptive Angular Replay (AAR), a method that mitigates the knowledge loss by applying feature normalization in hyperspherical space. Additionally, an adaptive scaling strategy (in task transitions) is proposed to improve the stability of the continual learning process. Experimental results demonstrate that AAR outperforms baseline methods, including cross-entropy (CE) and Nearest-Class-Mean (NCM) classifiers.

**Strengths:**

- This paper is easy to follow.
- The motivation is interesting.

**Weaknesses:**

1. I do not see any significant merit in the proposed method.
   - The method lacks novelty, as cosine similarity-based loss has already been explored in prior works (e.g., [1-2]).
   - The adaptive scaling strategy appears somewhat arbitrary. How are $s_{\text{min}}$ and $s_{\text{max}}$ selected? Does this strategy remain effective in other continual learning scenarios, such as online continual learning (OCL) and blurred boundary continual learning (BBCL)?
   - Furthermore, the paper does not include comparisons with recent continual learning methods, particularly those addressing the limitations of the cross-entropy classifier [3-5].
2. The experimental results are unconvincing.
   - The improvements over simple baselines, such as NCM and Angular, are marginal.
   - There is no analysis of the proposed method.
   - An ablation study of the proposed method is missing.
   - How do the learned representations change during task transitions compared to existing methods?
   - A more extensive evaluation under various scenarios would be beneficial, such as with fewer training iterations per task (e.g., in online continual learning), with a limited memory size, or with a large-scale dataset containing thousands of categories.

[1] Supervised Contrastive Replay: Revisiting the Nearest Class Mean Classifier in Online Class-Incremental Continual Learning \
[2] Co$^2$L: Contrastive Continual Learning \
[3] SS-IL: Separated Softmax for Incremental Learning \
[4] ScaIL: Classifier Weights Scaling for Class Incremental Learning \
[5] Mimicking the Oracle: An Initial Phase Decorrelation Approach for Class Incremental Learning

**Questions:**

Questions are mentioned in the Weaknesses section.

---

### Official Review · Reviewer_3N2V · 2024-11-08

**Soundness:** 2
**Presentation:** 1
**Contribution:** 1
**Rating:** 3
**Confidence:** 5

**Summary:**

The paper studies the ‘stability gap’ in continual learning.
It studies two questions in particular - 1) whether the past knowledge is lost abruptly or gradually when learning the new task, 2) and whether the past knowledge is completely lost at some point during the continual learning process.
The work finds that cross-entropy loss is responsible for the abrupt loss of information. It makes key observations showing that intermediate layers also undergo abrupt forgetting and shows that the class structure is significantly disturbed by a single gradient step during the continual learning process.
The paper proposes an approach called ‘ Adaptive Angular Replay’ to reduce abrupt forgetting between continual steps. The method includes the usage of normalized distance metric and adaptive scaling of softmax temperature.

**Strengths:**

The paper shows an interesting observation that the class structure is destroyed just by a single gradient step when learning the new task.
Other findings about abrupt performance loss are also quite interesting including that:
- that softmax based CE loss is responsible for abrupt performance loss,
- that the abrupt performance loss also occurs in intermediate layers along with the last fully-connected layer.

**Weaknesses:**

- The idea of normalizing the weight vector and feature vector to be 1 for continual learning is already proposed by Hou et al. 2019 [Learning a Unified Classifier Incrementally via Rebalancing]. Please describe how the proposed approach is different from the method by Hou et al.
- The paper is missing certain ablations and analysis to show how does the proposed approach resolves different identified limitations of softmax CE objective. Here are few open questions and missing analysis:
  - There is no ablation or analysis showing that normalized (angular) version of CE helps in reducing the abrupt forgetting problem.
  - The ACE baseline is not described. How is it different from the proposed AAR method?
  - Why does the proposed AAR method has larger benefits for the CLRS dataset? And Why does it not show improvements for Mini-ImageNet dataset.
  - Does AAR resolve the abrupt performance loss is intermediate layers as well?
  - TSNE in Figure 2 recoveres to original class structure at step: 7500. This observation is similar to Figure 6 TSNE plots. How do we know that the loss of class structure is the reason for overall worse continual learning performance?
- The conclusion to the second question '*Is past learned knowledge ever completely erased?*' raised in the abstract is not answered. Please clarify the conclusion.

**Questions:**

Please refer to the Weakness section for open questions.

**Suggestion**: An appropriate metric should be defined to quantify the abrupt loss in performance and compare different approaches. Per step performance curves and TSNE plots are not sufficient to show clear improvements.


**Other Comments**
- Angular ER term is used multiple times including abstract and introduction,  but not described anywhere.
- Table 1 not referenced anywhere.
- Line 067:  What is meant by ‘core network’? It is not defined anywhere.
- Typographical errors
   - Line 117: continua learning
   - Line 428: ‘ACE,ACE’ repeated twice
   - Line 070: grammatical error ‘network’s change dynamics’.
- There is not description about the Figure 1b in the manuscript.
- Incorrect reference to Fig 2 in Line 457
- Reference to Figure 6b is missing.

---

### Official Review · Reviewer_37M9 · 2024-11-09

**Soundness:** 1
**Presentation:** 1
**Contribution:** 2
**Rating:** 3
**Confidence:** 4

**Summary:**

* The paper studies the phenomenon of `stability gap` in continual learning which is the sharp decline of performance on old tasks during the initial stages of training on a new task.
* The paper conducts some analysis on why stability gap occurs within the network and where in the parameter space of the network the knowledge loss is maximum over the course of training on a new task.
* The paper then proposes a method to learn features in a hyperspherical space: Adaptive Angular Replay as a method to minimize the stability gap and thereby improve performance in general continual learning scenarios.

**Strengths:**

* The paper studies an interesting phenomenon: stability gap, which if properly understood and analysed will help the community design better continual learning algorithms.
* The proposed method can be interesting if properly analysed.

**Weaknesses:**

**Motivation**
* Given it’s the core problem being investigated in the paper, it can do a better job at explaining: a) what exactly is `stability gap` and b) why it is a critical problem and why it is important to study this problem in the overall context of continual learning.

**Analysis + Experiments**
* The empirical analyses presented in Section 3 are not convincing on the points/claims they are trying to make.

* First, both in Sections 3.1 and 3.2, the basic experimental setting (network, datasets, exact training hypers) are either missing partially or completely making the results and plots nearly inconclusive. Please specify exact details on these settings for the reader to understand and agree with the conclusions drawn.

* Second, insights in section 3.1 and 3.2 are not surprising.
* For instance in Section 3.1, since the training is end-to-end, it makes sense that the stability gap phenomenon can be attributed to internal feature changes (and not just last FC layer).
* In section 3.2, changes in earlier layers (layer 1 say) is less than changes in later layers (layer 5) can just be a result of the chain rule within backpropagation. I fail to understand why this is surprising.
* Finally, did you try ablations around training hyperparameters like optimizer and learning rate schedules? While optimizing we normally have a learning rate schedule which could be one of several types (step, cosine, linear, etc). This could have an impact on the stability gap phenomenon as well. Any insights on this front?

**Connection from Section 3 to Section 4**

* With the lack of an overall point from Section 3, I could not appreciate the method presented in Section 4.
* In Figure 4, the experimental details are not specified making the plot inconclusive in my mind.


**Presentation**

* The paper needs to be significantly more precise and specific in its wording. This is generally true for the entire text of the paper but I will provide some specific examples below.
* A specific example is that of Equation 3. The paper mentions Centered Kernel Alignment (CKA) but does not cite it [1]. Neither does it explain what HSIC (Hilbert-Schmidt Independence Criterion) is in Equation 3 giving the reader very little to go on.
* Is the method name Adaptive Angular Replay (mentioned in abstract and intro) or Adaptive Hyperspherical Replay (Section 4 title)?
* My above point on experimental settings missing from most analyses in the paper also falls under this point.


[1] Similarity of Neural Network Representations Revisited, arxiv:1905.00414

**Questions:**

Please see the Weaknesses section.

I would encourage the authors to re-write the paper making the motivation, analysis and conclusions drawn more solid. This requires a major re-writing of several sections within the paper.

---

### Note · Authors · 2024-11-14

I have read and agree with the venue's withdrawal policy on behalf of myself and my co-authors.